# Emerald Ash Borer Infestation-Induced Elevated Negative Correlations and Core Genera Shift in the Endophyte Community of *Fraxinus bungeana*

**DOI:** 10.3390/insects15070534

**Published:** 2024-07-14

**Authors:** Hua-Ling Wang, Zhen-Zhu Chen, Tuuli-Marjaana Koski, Bin Zhang, Xue-Fei Wang, Rui-Bo Zhang, Ruo-Qi Li, Shi-Xian Wang, Jian-Yong Zeng, Hui-Ping Li

**Affiliations:** 1College of Forestry, Hebei Agricultural University, Baoding 071001, China; 2Hebei Urban Forest Health Technology Innovation Center, Hebei Agricultural University, Baoding 071001, China; 3College of Life Sciences, Hebei University, Baoding 071002, China; 4Key Laboratory of Forest Germplasm Resources and Protection of Hebei Province, Hebei Agricultural University, Baoding 071001, China

**Keywords:** *Agrilus planipennis*, *Fraxinus bungeana*, endophyte, diversity

## Abstract

**Simple Summary:**

The emerald ash borer (EAB, *Agrilus planipennis* Fairmaire) is currently a highly destructive forest pest, causing substantial economic losses. Ash species (*Fraxinus* spp.) are economically invaluable components of natural forests and urban environments. However, these trees have been severely damaged by EAB. Endophytes, which are prevalent in plants, are hypothesized to contribute to the complex relationships between insects and plants. However, research on the effect of EAB infestation on the community changes in endophytic fungi and bacteria in phloem of *Fraxinus* is scarce. To bridge this gap, we compare changes in fungi and bacteria diversity, community makeup, and the potential roles of various endophytic fungi and bacteria in both EAB-infested and uninfected *Fraxinus bungeana* trees. We observed an elevation in bacterial richness, without notable changes in diversity, whereas fungal richness and diversity remained unaffected. Furthermore, we identified four key microbial genera undergoing substantial shifts post-infestation. The functional roles of endophytic fungi and bacteria also exhibited changes, with a decline in beneficial activities and an emergence of potentially detrimental functions. Network analyses have shown elevated negative correlations and core genera shift in EAB-infected phloem. The findings of our study contribute to a better understanding of the complex interactions between plants, insects, and endophytic microorganisms.

**Abstract:**

Endophytes, prevalent in plants, mediate plant–insect interactions. Nevertheless, our understanding of the key members of endophyte communities involved in inhibiting or assisting EAB infestation remains limited. Employing ITS and 16S rRNA high-throughput sequencing, along with network analysis techniques, we conducted a comprehensive investigation into the reaction of endophytic fungi and bacteria within *F. bungeana* phloem by comparing EAB-infested and uninfected samples. Our findings reveal that EAB infestation significantly impacts the endophytic communities, altering both their diversity and overall structure. Interestingly, both endophytic fungi and bacteria exhibited distinct patterns in response to the infestation. For instance, in the EAB-infested phloem, the fungi abundance remained unchanged, but diversity decreased significantly. Conversely, bacterial abundance increased, without significant diversity changes. The fungi community structure altered significantly, which was not observed in bacteria. The bacterial composition in the infested phloem underwent significant changes, characterized by a substantial decrease in beneficial species abundance, whereas the fungal composition remained largely unaffected. In network analysis, the endophytes in infested phloem exhibited a modular topology, demonstrating greater complexity due to an augmented number of network nodes, elevated negative correlations, and a core genera shift compared to those observed in healthy phloem. Our findings increase understanding of plant–insect–microorganism relationships, crucial for pest control, considering endophytic roles in plant defense.

## 1. Introduction

Plant endophytes, encompassing fungi, bacteria, and actinomycetes, reside within plant tissues throughout their life cycle, without inducing visible symptoms in the host [1]. These widely distributed endophytes have evolved intricate and specialized relationships with their hosts over time [2]. Endophytes have been demonstrated to offer various services to plants, ranging from exerting no significant impact or being harmful to their host, to enhancing plant defense mechanisms against environmental stressors [3,4,5,6]. For example, species related to *Pseudomonas* can inhibit the local defense mechanisms of plants [7,8], while nitrogen-fixing bacteria are renowned examples of microorganisms engaged in mutualistic relationships [9]. Therefore, alterations in the microbiome composition of endophytes could potentially produce significant impacts on the plant phenotype [10]. However, insect infestation can alter the microbial makeup of their host tree, as exemplified by certain bark- or wood-boring beetles and their corresponding host trees [11,12,13,14,15,16,17].

The emerald ash borer (EAB, *Agrilus planipennis* Fairmaire), belonging to the Buprestidae family of Coleoptera, *Agrilus* genus, poses significant harm to trees of the *Fraxinus* genus in the Oleaceae family [18]. It is currently the most destructive invasive forest pest in North America, as well as in Europe and Asia, causing significant economic losses [19,20,21]. This insect feeds on the phloem and outer xylem of *Fraxinus* species [22,23], forming characteristic S-shaped galleries, which disrupt water and nutrients transportation throughout the tree. This interruption leads to canopy thinning and ultimately, to the death of the tree within 1–3 years following the initial detection of symptoms [24]. Additionally, EAB infestation may trigger the occurrence of host canker diseases, accelerating tree mortality [24,25].

Ash species (*Fraxinus* spp.) are economically invaluable components of both natural forest ecosystems and the urban landscape [26]. For example, with over 8 billion forest ash trees in the United States alone, this genus holds an estimated worth of approximately USD 282.3 billion [27]. However, these trees have suffered immense devastation from EAB. In the US, EAB has caused widespread mortality among ash trees, earning the reputation of being the most destructive and financially burdensome wood-boring insect [28,29,30]. In China, the occurrence of EAB outbreaks persists, particularly in urban environments where EAB-induced damage escalated tenfold between 2010 and 2017 [31]. One ash species, *Fraxinus bungeana* DC., commonly known as Chinese ash, represents a distinguished member within the *Fraxinus* genus. Highly regarded for its ornamental landscaping value and timber cultivation potential, this species is unfortunately threatened by EAB [32].

Numerous investigations have revealed that EAB infestation in ash trees is significantly influenced by the complex interactions between trees, insects, and their microbiomes, thereby emphasizing the necessity of considering this tripartite relationship in managing EAB infestations [24,25]. Studies exploring the diversity of endophytic fungal communities in Chinese ash trees have uncovered patterns akin to those observed in other broad-leaved species, with Ascomycota and Basidiomycota being the dominant fungal phyla [33]. However, when *Fraxinus* species encounter external stressors, the balance of these fungal communities is disrupted, leading to significant shifts [34,35]. Ash dieback disease, caused by *Hymenoscyphus fraxineus*, serves as a prime example, significantly altering the endophytic fungal communities within Chinese ash (*Fraxinus excelsior* Linn.) [34]. These findings underscore the delicate nature of the interactions between fungal endophytes and their host trees during infestation by EAB, further emphasizing the importance of maintaining a balanced microbiome for the overall health and resilience of ash trees.

Currently, research efforts on endophytic fungi in Chinese ash primarily concentrate on analyzing fungal diversity shifts through high-throughput sequencing subsequent to pathogen invasion and also involve exploring fungal diversity within the phloem tissue affected by EAB, primarily relying on culturable methods of fungi identification [24,34,35]. However, there is a notable lack of research that simultaneously examines the impact of EAB infestation on the diversity of both endophytic fungi and bacteria residing in the phloem tissue of Chinese ash trees. This gap in knowledge represents a significant opportunity for future exploration, as a comprehensive understanding of these microbial communities is crucial for effective management of EAB infestations.

Hence, our objective is to compare the alterations in the endophytic fungi and bacteria populating the phloem tissue of *F. bungeana* trees under three distinct scenarios: trees impacted by EAB infestation (IP), uninfected tissue within infested trees (UP), and completely healthy, non-infested trees (CP). We endeavor to address the following questions: (1) Are there notable disparities in the endophytic communities of the phloem among these three groups? (2) Can we pinpoint specific endophytic taxa that exhibit a distinct response to EAB infestation? (3) What are the observed functional or endophytic interaction differences among these groups, and do healthy and unhealthy plants respond similarly to EAB infestation?

## 2. Materials and Methods

### 2.1. Collection and Processing of Phloem Tissue Samples

To collect phloem tissue samples affected by EAB infestation, *F. bungeana* trees lining the roadside in the Shunyi District of Beijing, China, underwent a careful selection process in late May 2021. These specimens were carefully chosen to ensure uniformity in age (10 years old), diameter at breast height (10–15 cm), and overall growth vigor, using dendrochronology for age determination. The sampled tissue consisted of the EAB-infected phloem (IP) extracted from the infested trees, along with the corresponding uninfected phloem (UP) and the control phloem (CP) obtained from healthy specimens.

The infestation status of the trees was confirmed through the observation of characteristic signs, including canopy thinning, bark splits, and/or D-shaped adult exit holes [18,29,36]. After identifying infested trees, we used a sterile surgical blade to carefully remove the bark. Following this procedure, a sterilized phloem punch (Ø ½ inch, disinfected with 70–95% ethanol between each use) was employed to carefully extract disks of infested phloem tissue (larval gallery) from various parts of the trunk at approximately breast height. The collected samples were then carefully placed into 1.5 mL sterile centrifuge tubes. Specifically, from each of the infected trees, three UP and three IP samples were collected from three distinct larval galleries, with one punch per gallery. The distance between the individual sampled trees varied from 7 to 10 m within each site. 

The identical methodology was applied in collecting uninfected phloem tissue samples. Healthy trees were identified by the absence of D-shaped adult exit holes, the presence of smooth bark, and no visible canopy thinning or epicormic shoots [18]. Samples were taken from the same height range as that used for the infested trees. The samples underwent rapid freezing in liquid nitrogen and were then promptly transported to the laboratory for storage at −80 °C until processing for DNA extraction. In total, we collected phloem samples from six sampled trees: three healthy trees each provided three CP (control/healthy) samples, while the remaining three EAB-infected trees each yielded six samples, with three designated as UP (uninfected) and three as IP (infested). This resulted in a total of twenty-seven samples. Each set of three samples (CP, IP, or UP) from the same tree were subsequently pooled separately to create a single replicate for each tree, resulting in nine samples ultimately utilized for the analysis after pooling.

### 2.2. DNA Extraction and Sequencing Analysis

Total DNA was extracted from phloem tissue samples using the E.Z.NA Soil DNA Kit (Omega Bio-Tek, Norcross, GA, USA), in accordance with the manufacturer’s instructions. For fungal PCR amplification, the primers ITS1F (5′-CTTGGTCATTTAGAGGAAGTAA-3′) and ITS2R (5′-GCTGCGTTCTTCATCGATGC-3′) were employed [37]. For bacterial PCR amplification, the primers 779F (5′-AACMGGATTAGATACCCKG-3′) and 1193R (5′-ACGTCATCCCCACCTTCC-3′) were used [38]. The PCR reaction mixture consisted of 4 μL 5× FastPfu Buffer, 2 μL 2.5 mM dNTPs, 0.8 μL each of forward and reverse primer, 0.4 μLFastPfu Polymerase, 10 ng template DNA, and supplemented with ddH_2_O to a final volume of 20 μL. PCR amplification conditions were as follows: initial denaturation at 95 °C for 5 min, followed by 30 cycles of denaturation at 95 °C for 30 s, annealing at 55 °C for 30 s, extension at 72 °C for 45 s, and a final extension at 72 °C for 10 min. Amplicons were extracted from 2% agarose gels using the AxyPrep DNA Gel Extraction Kit (Axygen Biosciences, Union City, CA, USA), following the manufacturer’s instructions. The PCR products were further purified using the AxyPrep DNA Gel Recovery Kit (Axygen Biosciences, Union City, CA, USA) and quantified using Qubit^®^3.0 (Life Technologies, Carlsbad, CA, USA). The genomic DNA library was prepared according to Illumina’s recommended protocol, specifically employing the Illumina Paired-End library construction method. Finally, the amplicon libraries were subjected to sequencing using paired-end (2 × 250) technology on the Illumina Novaseq 6000 platform [39], which was provided by Shanghai BIOZERON Co., Ltd. (Shanghai, China), in accordance with established protocols. In designing our sequencing analysis, we followed the methodologies outlined by Mogouong et al. (2021) [16] and Koski et al. (2024) [40], which did not include the use of positive and negative controls in the sequencing phase. It is important to note that we incorporated positive and negative controls during the PCR process to ensure the accuracy and reliability of our results.

### 2.3. Sequencing Data Analysis

The operational taxonomic units (OTUs) were clustered at a 97% similarity threshold utilizing UPARSE 7.1. Subsequently, OTUs were taxonomically classified using the RDP Classifier and Bayesian algorithm from the UNITE database, employing a confidence threshold of 0.7. Alpha diversity analysis was conducted separately for endophytic bacteria and fungi utilizing Mothur software (v.1.43.0) [41], including metrics such as the Chao1 index, ACE index, Shannon index, and Simpson index. To assess disparities in the endophytic bacterial and fungal community structure between uninfected and infested phloem of *F. bungeana* trees, Beta diversity analysis was performed, using principal coordinate analysis (PCoA), based on Bray–Curtis distances at the genus level, on nine samples (three UP, IP, and CP samples). Additionally, analysis of similarities (ANOSIM) was employed to further assess the distinctions between groups [12]. To investigate significant differences in the microbial composition among groups, a single-factor analysis of variance was conducted using SPSS software (v.29.0).

A functional prediction analysis of endophytic fungi was executed utilizing FunGuild and FAPROTAX databases on the Shanghai LingboMicroclass Cloud Platform, with heatmaps generated to depict the top 20 sub-functions, based on relative abundance. Pearson correlation analysis was then performed on the top 50 abundant fungal and bacterial genera, with data exhibiting correlation coefficients exceeding 0.6 and *p*-values below 0.05 selected for network construction utilizing Gephi software (v.0.10.1) [42].

## 3. Results

### 3.1. Alpha Diversity of Endophytic Fungal and Bacterial Communities

The differences in the alpha diversity indices of endophytic fungi and bacteria in the phloem of *F. bungeana* trees among three groups were analyzed using one-way analysis of variance (ANOVA) (Table 1). All samples from the phloem of *F. bungeana* trees demonstrated a sequencing coverage of over 99%, indicating that the sequencing data accurately reflected the diversity of endophytic fungal and bacterial communities within the samples.

Fungal ITS sequencing results revealed a significantly higher Simpson index in the UP group compared to the CP group, with no significant difference between the IP and CP groups. All three groups were similar in terms of OTU abundance (Appendix A), Chao1 index, ACE index, and Shannon index. In bacterial 16S sequencing, the IP group exhibited a higher OTU abundance (Appendix A), Chao1, and ACE indices compared to the results for the CP and UP groups. However, the UP and CP groups were similar in regards to these indices. No significant variations were noted regarding the Shannon and Simpson indices among the three groups.

In the infested samples, the fungi richness (Chao 1, ACE) and diversity (Shannon and Simpson indexes) in the IP group remained largely unaffected, whereas bacterial richness significantly increased, without affecting its diversity. In the UP group, only fungal diversity showed a significant decrease; bacterial richness and diversity, as well as fungal richness, remained unaffected.

### 3.2. β Diversity of Endophytic Bacterial and Fungal Communities

Principal coordinate analysis (PCoA) indicated that the endophytic fungi present in the phloem of *F. bungeana* explained 26% and 22% of the variance in PC1 and PC2, respectively. It is worth noting that the UP and CP groups exhibited overlapping positions on the PCoA plot, suggesting a comparable community structure. In contrast, the composition of the endophytic fungal communities in the IP group differed significantly (R = 0.506, *p* = 0.014) in both the UP and CP groups (Figure 1A).

The endophytic bacteria residing in the phloem of *F. bungeana* accounted for 68% and 31% of the variation in PC1 and PC2, respectively. Substantial overlap was observed between the UP and CP groups, suggesting a similar endophytic bacterial community structure. Nevertheless, slight variations (R = 0.300, *p* = 0.069) were detected in the composition of endophytic bacterial communities within the IP group when compared with the results for both the UP and CP groups (Figure 1B).

### 3.3. Impact of EAB Infestation on Endophytic Fungal Composition

#### 3.3.1. Dominant Fungal Community Composition

Across the three groups studied, a total of four fungal phyla were identified in the phloem of *F. bungeana*. Among these, Ascomycota and Basidiomycota were found to be the prevalent phyla, both demonstrating relative abundances exceeding 1%. Notably, Ascomycota was the most abundant phylum in all three groups, accounting for 76.81%, 96.60%, and 88.00% of the relative abundances in the IP group, the UP group, and the CP group, respectively (Figure 2A).

An extensive annotation of endophytic fungi at the genus level in the phloem of *F. bungeana* revealed the presence of 395 genera (Appendix A) across all three groups. Significant changes in the dominant fungal genera were observed in the IP group. Specifically, in the CP group, *Penicillium* (29.50%), *Bradymyces* (12.84%), and unclassified fungi (9.11%) were the three most abundant genera. However, in the IP group, the three dominant genera shifted to *Botryosphaeria* (11.98%), unclassified fungi (10.18%), and *Cyberlindnera* (9.94%). In contrast, the relative abundances of *Penicillium* and *Bradymyces* decreased to 0.20% and 13.86%, respectively, in the IP group. Meanwhile, in the UP group, the prevalent genera were identified as *Medicopsis* (20.84%), *Diaporthe* (14.18%), and *Preussia* (13.95%). It is worth noting that the relative abundance of *Penicillium* was lower, at 3.08%, in the UP group, while *Bradymyces* was significantly higher, at 13.86% (Figure 2B).

#### 3.3.2. Dominant Bacterial Community Composition

At the phylum level, we detected a total of 14 phyla of endophytic bacteria in the phloem of *F. bungeana* across three groups, with the dominant phylum identified as Proteobacteria (with relative abundances > 1%). Proteobacteria consistently prevailed, constituting 99.03%, 99.44%, and 99.62% of the relative abundance in the IP, UP, and CP groups, respectively (Figure 3A).

Examining the genus level revealed distinct profiles among the dominant genera in the *F. bungeana* sample groups. The top three dominant genera across these groups were *Burkholderia-Caballeronia-Paraburkholderia* (46.12%, 46.75%, 46.45%), *Bradyrhizobium* (40.87%, 2.10%, 34.92%), and *Sphingomonas* (8.72%, 44.22%, 14.23%), respectively. Notably, in the IP group, *Sphingomonas* was significantly higher in abundance compared to the results for the healthy CP group, resulting in its elevation to the second position, whereas *Bradyrhizobium* experienced a decrease, leading to its descent to the third rank. In the UP group, despite minor fluctuations in the abundance of *Bradyrhizobium* and *Sphingomonas*, the rankings of these genera remained unchanged (Figure 3B).

### 3.4. Differential Endophytic Fungal and Bacterial Communities

To further explore endophytic fungi and bacteria involved in the host–insect interaction, ANOVA analysis was conducted on the endophytic microbes present at abundances exceeding 1% in samples across the three groups. In the IP group, a unique fungal genus, *Medicopsis*, was discovered in the phloem of *F. bungeana*. Its abundance remained similar in the IP and CP groups but was notably elevated in the UP group versus the CP group (Figure 4A). For endophytic bacteria, three distinct genera were identified: *Bradyrhizobium*, *Rhodopseudomonas*, and *Pseudomonas* (Figure 4B–D). Notably, the abundances of *Bradyrhizobium* and *Rhodopseudomonas* were significantly reduced in the IP group compared to the results for the CP group, whereas *Pseudomonas* abundance was significantly increased in the IP group.

### 3.5. Functional Profiles of Endophytic Fungal and Bacterial Communities

#### 3.5.1. Functional Changes in Endophytic Fungal Communities

Across the CP, IP, and UP examined groups, the primary fungal functions encompassed undefined saprotroph (47.34%, 17.69%, and 12.21%, respectively), animal pathogen (4.92%, 10.69%, and 22.16%, respectively), endophyte–plant pathogen, plant pathogen, plant pathogen–wood saprotroph, and dung saprotroph–plant saprotroph, among others. Specifically, among functional groups with abundances exceeding 1%, the undefined saprotroph and animal pathogen groups exhibited notable shifts post-EAB infestation. The abundance of undefined saprotrophs decreased significantly in the IP group compared to the CP group, whereas the abundance of animal pathogens increased markedly in the UP group relative to both the CP and IP groups (Figure 5A). The fungal taxa attributed to undefined saprotrophs comprised *Paraconiothyrium* and *Capnobotryella*, while those linked to animal pathogens included *Medicopsis* and *Nigrograna*.

#### 3.5.2. Functional Changes in Endophytic Bacterial Communities

Across the CP, IP, and UP detected groups, the primary bacterial functions comprised chemoheterotrophy (34.38%, 43.75%, and 35.29%, respectively), aerobic chemoheterotrophy (34.37%, 43.35%, and 35.27%, respectively), and nitrogen fixation (26.19%, 1.82%, and 23.16%, respectively). Post-EAB infestation, the IP group exhibited a significant reductions in nitrogen fixation, phototrophy, and photoheterotrophy compared to the results for the UP and CP groups. Conversely, the abundance of plant pathogens increased significantly in the IP group relative to the other two groups. The genus *Bradyrhizobium* was predominantly associated with nitrogen fixation, whereas *Rhodopseudomonas* was the primary genus linked to both phototrophy and photoheterotrophy. Furthermore, the bacteria primarily responsible for the plant pathogen function belonged predominantly to the genus *Pseudomonas* (Figure 5B).

### 3.6. Co-Linear Network

To explore microbial interactions, we constructed a co-linearity network using the 50 most abundant fungal and bacterial genera, based on Pearson correlation analysis. In the CP group, the network features 46 nodes (predominantly fungal: 93.48%) and 102 edges, with a strong positive correlation (85.29%). In contrast, the network of the IP group shows 49 nodes (73.47% fungal) and 86 edges, with a slightly lower positive correlation (79.07%). Meanwhile, the network of the UP group has 45 nodes (93.33% fungal) and 74 edges, again with a strong positive correlation (78.38%). Notably, in the CP group, fungi dominate the endophytic community, indicating robust positive interactions (Figure 6A). However, in the EAB-infested phloem (IP group), there is a notable increase in bacterial nodes and a rise in negative correlations (Figure 6B), suggesting that EAB infestation disrupts the bacterial community balance, leading to intensified fungal–bacterial interactions among endophytic fungi and bacteria. The microbial network of the UP group is similar to that of the CP group (Figure 6C).

From a network composition standpoint, endophytic bacteria are primarily proteobacteria, whereas endophytic fungi are predominantly Ascomycota. In the IP group, the core genera decrease. In the CP groups, there are 10 interconnected core genera, including *Medicopsis*, *Nigrograna*, *Kwoniella*, *Fragosphaeria*, *Truncatella*, *Lulworthia*, *Kernia*, *Mortierella*, *Phoma*, and *Querciphoma*, with *Medicopsis*, *Nigrograna*, and *Kwoniella* being dominant (Figure 6A). However, in the IP groups, the core genera undergo a shift, with new genera like *Cyberlindnera*, *Diaporthe*, an unclassified genus, *Fusarium*, *Aspergillus*, *Humicola*, *Lodderomyces*, *Chaetomium*, and *Trametes* emerging as dominant (Figure 6B). Among these, *Cyberlindnera*, *Diaporthe*, the unclassified genus, *Fusarium*, and *Aspergillus* emerge as the dominant genera. Additionally, within infested trees in the UP groups, the core genera comprise *Aspergillus*, *Parathyridaria*, *Orbilia*, *Phialemonium*, *Thielavia*, *Papulaspora*, and *Ochrocladosporium*, with *Aspergillus* as the only dominant genus (Figure 6C). Notably, *Medicopsis*, *Nigrograna*, *Kwoniella*, and *Aspergillus* are consistently present in the co-linearity networks of all three groups. In the IP group, *Aspergillus* transitions from being peripheral to being core, while *Medicopsis*, *Nigrograna*, and *Kwoniella* show the opposite trend.

## 4. Discussion

Endophytic fungi and bacteria are crucial for plant health, and insect infestations can significantly alter the composition of endophytic microbial communities [24]. Nonetheless, there is a lack of understanding about how EAB affects the endophytic fungi and bacteria present in *F*. *bungeana.* We found that infested trees exhibited stable endophytic fungal OTUs and richness compared to healthy trees, yet their diversity was significantly reduced. Conversely, while endophytic bacterial diversity remained unchanged, there was a notable increase in bacterial richness and OTUs. The observed variations in diversity and abundance between fungi and bacteria are potentially due to their differing responses to EAB infestation [43]. Likewise, PcoA analyses reveal a significant impact of EAB infestation on the structure of endophytic fungal communities within infected host tissues, whereas its effect on uninfected tissues is minimal. Slight variations in endophytic bacterial communities in the infested group compared to uninfected and control groups suggest minimal EAB influence on bacterial structure, but hint at possible ongoing changes in the bacterial community of affected trees.

This observation aligns with the findings of Hao et al. (2022) [44], who noted no significant difference in fungi richness between diseased and healthy *Pinus massoniana* trees infested by a plant pathogenic nematode *Bursaphelenchus xylophilus* (pinewood nematode, PWN), but there was a significant difference in the fungal diversity index. The infection of PWN significantly impacted both the endophytic fungal and bacterial communities of *P. massoniana* branches and trunks. This finding concurs with our fungal results but diverges from our bacterial findings. Our results conflict with those of Zhang (2022) [45], who found that the infestation of *Ambrosia artemisiifolia* L. by *Spodoptera litura* Fabricius led to a significant decrease in the diversity of endophytic fungi in the roots, while having no significant impact on the richness and diversity of endophytic bacteria. The observed discrepancies between our results and those of Hao et al. (2022) [44] and Zhang (2022) [45] could be attributed to variations in experimental materials, conditions, infestation severity, and methodological differences.

In our study, Ascomycota and Basidiomycota were identified as the dominant fungal phyla in all three tested groups, which aligns with and reinforces the results obtained by Held et al. (2021) [24] and Agostinelli et al. (2021) [33] in their respective studies on endophytic fungi associated with the genus *Fraxinus*. Proteobacteria, the most predominant phylum frequently isolated from plants [46], dominated the bacterial community. However, its relative abundance decreased in the infested sample, corroborating observations made during the infestation of *Pinus koraiensis* by *Monochamus alternatus* Hope [14]. *Alternaria*, *Fusarium*, and *Penicillium*, which are present in all tested samples and commonly found in European ash trees as well as in EAB galleries of North American green ash trees, may represent a general characteristic of ash trees [24,47,48]. *Penicillium* (29.50%) was found to be the most prevalent in the CP group compared to the results for the UP and IP groups, which might be attributed to the fact that numerous species belonging to the *Penicillium* genus are recognized as plant endophytes and are renowned as abundant sources of bioactive secondary metabolites [49]. Hence, it is justifiable to detect a significant presence of the genus in healthy trees. The domination of *Botryosphaeria* and *Cyberlindnera* in the IP group might be explained by the fact that *Botryosphaeria* fungi are typically opportunistic pathogens [50], while *Cyberlindnera* fungi have been proven to exhibit antagonistic activities against a range of pathogenic and saprophytic filamentous fungi, making them beneficial in fighting EAB infestation [51]. The prevalence of *Medicopsis* (20.84%) and *Diaporthe* (14.18%) in the UP group might be due to the fact that some species of *Medicopsis* have been proven to be conditional pathogens [52], while *Diaporthe* occur in large numbers on stems of the dying ash trees (*Fraxinus excelsior*) [47]. We postulate that, in addition to directly affecting the encountered segments of the tree, EAB infestation also indirectly impacts other parts of the tree, ultimately leading to the death of the ash trees.

Interestingly, the taxonomic composition bar plots show that the relative abundance of *Sphingomonas* increased, while that of *Bradyrhizobium* decreased in the IP group compared to the CP group. *Sphingomonas* has been reported to enhance plant growth under stress conditions [53], whereas *Bradyrhizobium* is known for its intimate symbiotic association with leguminous plants [54]. Despite the fact that both are beneficial bacteria, they exhibited contrasting trends in the IP group versus the CP group. This suggests that they may be involved in the host–insect interaction through different mechanisms. The increase in *Sphingomonas* abundance could indicate a role in strengthening the defense of the plant against insect infestation or improving plant health, while the decrease in *Bradyrhizobium* might reflect a change in the symbiotic relationships or nutritional status of the plant, affecting insect behavior or survival. These findings highlight the complexity of microbial communities and their potential impact on plant–insect interactions.

In the EAB-infested group, four genera, namely *Medicopsis*, *Bradyrhizobium*, *Rhodopseudomonas*, and *Pseudomonas*, underwent significant shifts, with varying trends across the tested groups. This suggests that these genera may be implicated in the process of EAB infestation, or alternatively, that they could be opportunists exploiting the potential ecological niches created by EAB infestation [55]. The relative abundance of functional endophytic microbial communities within the phloem of *F. bungeana* trees undergoes significant shifts. These alterations subsequently impact microbial functions, leading to a notable increase in harmful functions such as plant pathogenesis, for both fungi and bacteria, and a decrease in beneficial functions for bacteria, like nitrogen fixation, phototrophy, and photoheterotrophy. The fungal taxa linked to the specific functional changes were identified as *Paraconiothyrium*, *Capnobotryella Medicopsis* and *Nigrograna*, thus highlighting the significant impact of EAB infestation on the composition of fungal communities within these specific functional categories. A study analyzing fungi isolated from EAB galleries in damaged *Fraxinus pennsylvanica* trees in the US observed a similar increase in harmful fungal functions, specifically noting a significant prevalence of canker pathogens [24]. Some of these pathogens were found to induce canker diseases of varying severity in their hosts [25], suggesting that canker fungi play a critical role in weakening ash trees and may potentially contribute to their death following EAB infestation. Specifically, the endophytic fungus *Medicopsis* experiences a substantial surge in abundance in the UP group, indicating its sensitivity to EAB infestation. 

Interestingly, Bains et al. (2020) [52] have identified *Medicopsis romeroi*, isolated from *Suaeda monoica* mangrove stems, as a conditional pathogen through in vitro pathogenicity tests. Aligning with this finding, our fungal functional prediction within the phloem of *F. bungeana* categorizes *Medicopsis* as an animal pathogen. Additionally, remarkable changes in animal pathogenic fungi are observed in the IP group, possibly indicating a plant immune response to insect pests. This observation concurs with the results of previous research, suggesting that the endophytic microbiota of certain plants harbors microorganisms that are detrimental to the growth and development of insects [56,57]. For instance, Wang et al. (2019) [56] found that inoculating the endophytic fungus *Ophiostoma minus* into pine sections invaded by *Sirex noctilio* larvae significantly increased larval mortality. Similarly, Qin et al. (2021) [57] inoculated maize seeds with endophytic fungi *Beauveria* sp. YC1 and *Alternaria* sp. GX5, discovering that the resulting maize ears were unfavorable for the infestation of the fall armyworm (*Spodoptera frugiperda*), thereby significantly reducing its reproductive rate. These findings not only underscore the significance of endophytic fungi in plant–insect interactions, but also unveil the potential application of these pathogenic fungi in biological control strategies.

Bacterial functional predictions reveal substantial decreases in beneficial functions, such as nitrogen fixation [58], phototrophy, and photoheterotrophy [59], paralleled by an increase in harmful functions, such as plant pathogens [60], within the EAB infested phloem of *F. bungeana*. Through identifying particular bacterial variations that contribute to functional alterations, it was discovered that *Bradyrhizobium*, a widespread nitrogen-fixing bacterium, fosters plant growth by fixing nitrogen, synthesizing substances that stimulate growth, and bolstering plant resistance [61]. However, its population declines notably in the IP group, which may diminish the nitrogen fixation ability of *F. bungeana*. *Rhodopseudomonas*, a key photosynthetic bacterium, synthesizes crucial nutrients for plant development, releases growth-promoting factors, enhances photosynthetic efficiency, as well as produces antiviral and antifungal proteins, aiding plant defense [62,63]. However, a notable decline in *Rhodopseudomonas* abundance post-infestation could negatively impact the photosynthetic capabilities and overall health of the *F. bungeana*. To maintain plant health, it is imperative to comprehend and cultivate the microbial communities present within the plant, with a specific focus on beneficial bacteria such as *Rhodopseudomonas*. Future research endeavors may focus on developing strategies to enhance the proliferation and robustness of these beneficial microorganisms, ultimately enhancing plant vigor and productivity. We observed a notable increase in the population of *Pseudomonas* bacteria, with some strains capable of inducing cankers [64], in the infested plant sections. This surge could possibly be attributed to EAB harboring associated microbes or to environmental microbes infiltrating through infestation wounds [24]. Therefore, it is hypothesized that infestation by EAB weakens *F. bungeana* trees, thereby enabling an invasion or prevalence of *Pseudomonas*. Consequently, this may trigger stem diseases in *F. bungeana*, which could potentially hasten the deterioration and demise of these trees.

The co-linearity network within the healthy phloem of *F. bungeana* trees demonstrates remarkable stability, primarily attributed to the preponderance of positive correlations among diverse genera and synergistic fungi–bacteria interactions [65]. However, this balance is disrupted in the IP group, giving rise to an increase in negative correlations and core genera shifts compared to those observed in healthy phloem. The results could be explained by the involvement of microbiota in these connections, which might prime the immune system of the plant for the accelerated activation of defense mechanisms against invading pathogens [66,67,68].This result also aligns with previous research indicating that endophytes in infected plants exhibit a modular topology, increased network complexity, a significant rise in negative correlations, and intensified fungal–bacterial interactions compared to their counterparts in healthy plants [43,65]. Genera that might be crucial in maintaining the microecological balance of the plant host were identified. For example, *Medicopsis*, *Nigrograna*, and *Kwoniella*, widely recognized as plant probiotics [52,69], exhibiting co-occurrence in the three tested networks, undergo a significant role transition from being core microbiota to occupying peripheral roles. This shift could be intricately linked to the resilience and ability of *F. bungeana* to withstand the infestation of EAB [52,69]. Similarly, *Aspergillus* also occurred simultaneously in the three tested networks, but experienced a transition from being a peripheral genus to occupying a core position upon EAB infestation. This transition is potentially attributed to its antagonistic properties against plant pathogens and its insecticidal abilities [70,71]. Meanwhile, as plant probiotics shift from being core microbiota to playing peripheral roles, notable plant pathogens like *Diaporthe* and *Fusarium* emerge as core genera in the infested group. This suggests that after repeated interactions between pathogenic and endophytic bacteria, the pathogens eventually prevail, reproducing prolifically, while the beneficial bacteria dwindle significantly [72].

## 5. Conclusions

Compared to uninfected phloem, the microbial diversity and community structure of *F. bungeana* infested with EAB undergo considerable alterations. Consequently, this shift results in a notable increase in and proliferation of harmful genera, such as *Pseudomonas*, renowned for its ability to cause cankers, and a substantial decline in beneficial genera such as *Bradyrhizobium* and *Rhodopseudomonas*, which are recognized for their roles in nitrogen fixation and photosynthesis. Consequently, crucial functions like nitrogen fixation and phototrophy are reduced. In network analysis, the endophytes in infested phloem exhibited a modular topology, an elevated negative correlation, and a core genera shift compared to those observed in healthy phloem. Future research efforts should focus on incorporating cultivation techniques to effectively isolate and accurately identify individual species within the endophytic microbial community. This will allow for a more in-depth exploration of their specific roles in the infestation process initiated by EAB, ultimately enhancing our understanding of the intricate interactions between plant endophytes and insect–plant associations. Such investigations hold promise for uncovering mechanisms underlying insect infestation and may contribute to the development of targeted strategies for pest management and plant protection.

## Figures and Tables

**Figure 1 insects-15-00534-f001:**
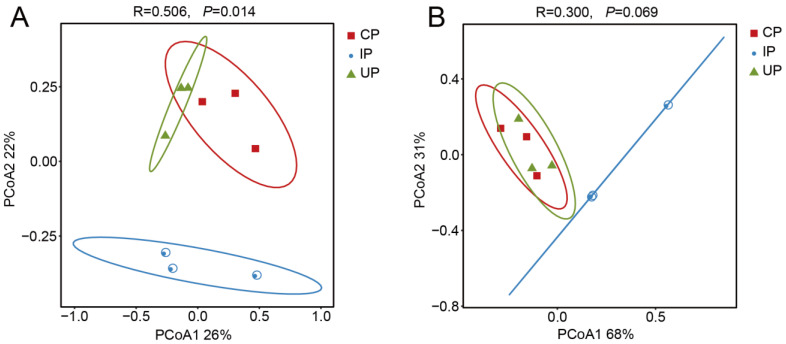
Principal coordinate analysis of endophytic fungi (**A**) and bacteria (**B**) in phloem of *F. bungeana*.

**Figure 2 insects-15-00534-f002:**
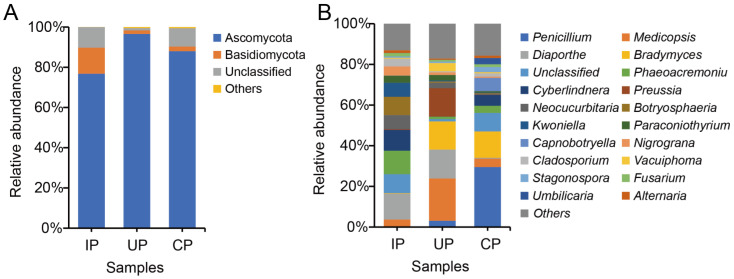
Community structure of endophytic fungi from the phloem in *F. bungeana* at the phylum (**A**) and genus (**B**) levels. In figure (**A**), the category “Others” represents the aggregated relative abundance of each fungal taxa with a relative abundance of less than 1%. In Figure (**B**), “Others” refers to the collective abundance of the fungal taxa grouped within the lowest quintile of their ranked abundance.

**Figure 3 insects-15-00534-f003:**
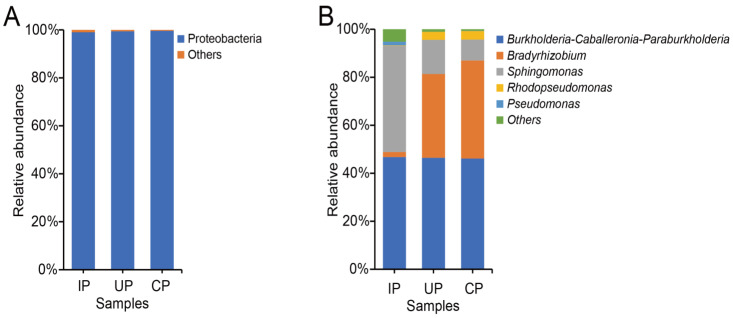
Community structure of endophytic bacteria from the phloem in *F. bungeana* at the phylum (**A**) and genus (**B**) levels. In both figures, the category “Others” represents the aggregated relative abundance of each bacterial taxa constituting less than 1% of the total community.

**Figure 4 insects-15-00534-f004:**
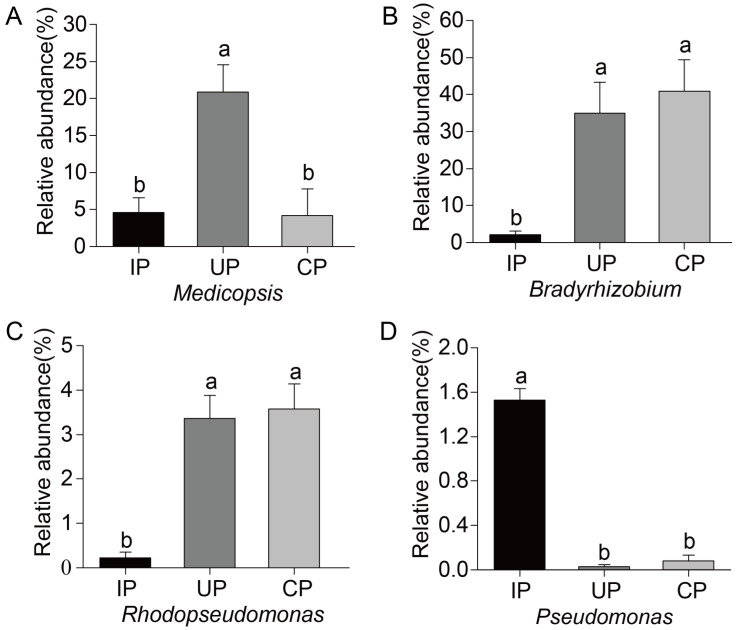
Differential genus of endophytic bacteria in the phloem of *F. bungeana.* (**A**) *Medicosis*; (**B**) *Bradyrhizobium*; (**C**) *Rhodopseudomonas*; (**D**) *Psedomonas*. a, b indicated significant difference among samples at the *p* < 0.05 level.

**Figure 5 insects-15-00534-f005:**
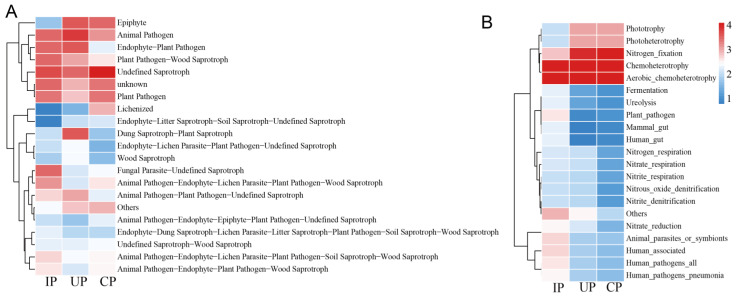
Functional prediction of endophytic fungi (**A**) and endophytic bacteria (**B**) in the phloem of *F. bungeana.* In both figures, the category “Others” refers to the collective abundance of fungal taxa grouped within the lowest quintile of their ranked abundance.

**Figure 6 insects-15-00534-f006:**
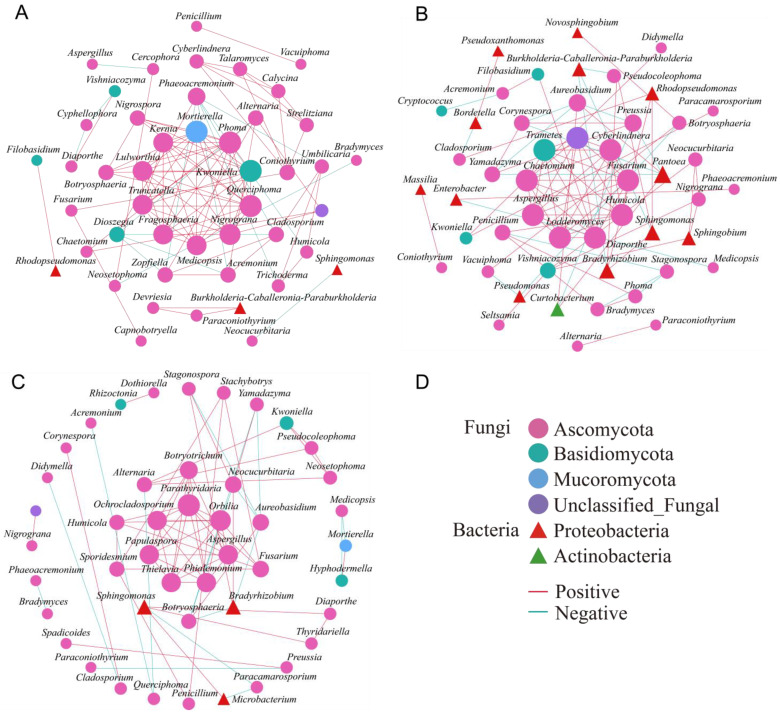
Correlation network analysis of endophytes in *F. bungeana* at the genus level. (**A**–**C**) represent control phloem (CP), infested phloem (IP), and uninfected phloem (UP), respectively. Circles depict endophytic fungi, while triangles represent endophytic bacteria. (**D**) Different colors denote distinct phyla. The sizes of the shapes indicate the degree of connectivity, and the lines connecting the points signify the relevance between them. Specifically, the red lines represent a positive correlation, whereas the blue lines indicate a negative correlation.

**Table 1 insects-15-00534-t001:** Separate analysis of alpha diversity indices for endophytic fungi and bacteria in the phloem of *Fraxinus bungeana*.

	Endophytic Fungi			Endophytic Bacteria		
	IP	UP	CP	IP	UP	CP
OTU	573.67 ± 171.66 a	441.33 ± 82.07 a	646.33 ± 20.80 a	242.00 ± 12.49 a	89.67 ± 4.91 b	81.00 ± 12.74 b
Chao1	705.16 ± 115.12 a	724.98 ± 91.09 a	863.35 ± 96.71 a	340.31 ± 35.36 a	118.96 ± 3.49 b	128.44 ± 33.20 b
ACE	708.91 ± 116.98 a	743.53 ± 77.49 a	885.41 ± 94.11 a	346.15 ± 29.12 a	123.13 ± 6.96 b	131.81 ± 31.13 b
Shannon	4.95 ± 0.72 a	4.30 ± 0.75 a	6.31 ± 0.32 a	2.26 ± 0.33 a	2.35 ± 0.05 a	2.18 ± 0.15 a
Simpson	0.11 ± 0.01 ab	0.17 ± 0.05 a	0.04 ± 0.01 b	0.43 ± 0.11 a	0.28 ± 0.02 a	0.31 ± 0.04 a
Coverage	0.997 ± 0.00 a	0.997 ± 0.00 a	0.997 ± 0.00 a	0.997 ± 0.00 a	0.999 ± 0.00 a	0.999 ± 0.00 a

Note: The data in the table reflect means ± SE; different letters in the same column indicate significant difference among samples at the *p* < 0.05 level. IP, UP, and CP represent infested phloem, uninfested phloem, and control phloem, respectively.

## Data Availability

The raw sequence data were deposited in the National Center for Biotechnology Information (NCBI) Sequence-Read Archive (SRA) database under accession number PRJNA1101710.

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
