# Peer review of "Emerald Ash Borer Infestation-Induced Elevated Negative Correlations and Core Genera Shift in the Endophyte Community of Fraxinus bungeana"

_insects, 2024, doi:10.3390/insects15070534_

Round 1

Reviewer 1 Report

Comments and Suggestions for Authors

Summary: The study presently titled "Emerald ash borer infestation-induced elevated negative correlations and core genera shift in endophyte community of Fraxinus bungeana" explored the endophytic microbiome of F. bungeana phloem infested with emerald ash borer. The study determined that there were significant differences in infested and uninfested phloem tissue, but those differences varied among the fungal and bacterial communities. There were differences between the functional groups and microbial genera present in each sample, which could impact the total microbiome. The study addresses an important knowledge gap on understanding how the microbiome of ash trees are affected by EAB, which could lead to other studies that explore how the microbiome influences the susceptibility of ash to EAB. The study design was well-developed.

Genera Comments: One overall suggestion is for the authors to remove concluding or interpretive statements from the results sections as these should only be included in the Discussion section. Many times the authors ended their results with such a statement. I suggest the authors also revise their title as it is confusing and clunky. Finally, all images in the manuscript were blurry, so that should be addressed prior to publication. My greatest issue with the manuscript is how the authors use "before" and "after" EAB infestation to describe the results. These words do not align with their study. Specific comments are below:

Line 119: Please state the DBH used. How did you determine uniformity in age?

Section 2.1: How many trees were sampled and how many total samples were used for the analysis after pooling?

Section 2.2: Were positive and negative controls included in the sequencing analysis? This needs to be noted. 

Lines 162-163: It is not clear if the alpha diversity analyses were conducted on bacterial and fungal groups separately or all together. This should be made clear in this section and in Table 1.

Lines 164-167: How disparities were assessed before and after EAB infestation needs to be made more clear. What samples were used? How was "before" and "after" determined? This seems like it should have been separate experiment with EAB infestation performed. Based on the current samples mentioned, I do not believe you can make the case that the results represent before and after. The described samples represent infested, uninfected, and healthy.  

Lines 378-380: This sentence is confusing and needs reworded. 

Lines 490-492: This conclusion is a bit bold for me. While you demonstrated that there are changes in the endophytic composition between EAB infested and uninfested phloem, you did not show that antagonistic relationships increased only that those functional groups have increased. The conclusion should be reworded to avoid this overstatement. 

Table 1, Note: This note is confusing I believe due to a spelling error. Please revise.

Comments on the Quality of English Language

Overall, the quality of the writing is acceptable yet there were many typos and spellings error. Thus, the authors should review the paper and make corrections. In addition, the authors alternate between saying "the EAB" and "EAB." I recommend the authors use EAB without the in front of it. 

Reviewer 2 Report

Comments and Suggestions for Authors

The manuscript is nicely structured. All the claims are backed up by detailed graphs and figures.

In the introduction, the significance of this pest is only described from North America. Data from Europe and Asia needs to be added.

Why were the species not identified to the species level, but to higher classification categories? The species list should be added as a supplementary file if you identified the species to that level. A list of all the genera which you mention (395) should be added in the form of a table with them categorized by the IP, UP, as a supplementary file. Sorry for the delay in the review process.

All the best,

Reviewer

Comments on the Quality of English Language

The English is fine, only a few words should be changed.
